# Human exposure to zoonotic malaria vectors in village, farm and forest habitats in Sabah, Malaysian Borneo

Rebecca Brown[1]*, Tock H. Chua[2], Kimberly Fornace[3‡], Chris Drakeley[3‡], Indra Vythilingam[4‡], Heather M. Ferguson[1]

1 Institute of Biodiversity, Animal Health and Comparative Medicine, College of Medical, Veterinary and Life Sciences, University of Glasgow, Glasgow, United Kingdom, 2 Department of Pathobiology and Medical Diagnostics, Faculty of Medicine and Health Sciences, Universiti Malaysia Sabah, Kota Kinabalu, Sabah, Malaysia, 3 Faculty of Infectious and Tropical Diseases, London School of Hygiene and Tropical Medicine, London, United Kingdom, 4 Department of Parasitology, Faculty of Medicine, Kuala Lumpur, University of Malaya, Malaysia

☯ These authors contributed equally to this work.
‡ KF, CD and IV also contributed equally to this work.
* rebbrown552@gmail.com

**Data Availability Statement:** The data underlying the results presented in the study are available from Harvard dataverse: https://doi.org/10.7910/DVN/3QG1HP.

## Abstract

The zoonotic malaria parasite, *Plasmodium knowlesi*, is now a substantial public health problem in Malaysian Borneo. Current understanding of *P. knowlesi* vector bionomics and ecology in Sabah comes from a few studies near the epicentre of human cases in one district, Kudat. These have incriminated *Anopheles balabacensis* as the primary vector, and suggest that human exposure to vector biting is peri-domestic as well as in forest environments. To address the limited understanding of vector ecology and human exposure risk outside of Kudat, we performed wider scale surveillance across four districts in Sabah with confirmed transmission to investigate spatial heterogeneity in vector abundance, diversity and infection rate. Entomological surveillance was carried out six months after a cross-sectional survey of *P. knowlesi* prevalence in humans throughout the study area; providing an opportunity to investigate associations between entomological indicators and infection. Human-landing catches were performed in peri-domestic, farm and forest sites in 11 villages (3–4 per district) and paired with estimates of human *P. knowlesi* exposure based on sero-prevalence. *Anopheles balabacensis* was present in all districts but only 6/11 villages. The mean density of *An. balabacensis* was relatively low, but significantly higher in farm (0.094/night) and forest (0.082/night) than peri-domestic areas (0.007/night). Only one *An. balabacensis* (n = 32) was infected with *P. knowlesi*. *Plasmodium knowlesi* sero-positivity in people was not associated with *An. balabacensis* density at the village-level however post hoc analyses indicated the study had limited power to detect a statistical association due low vector density. Wider scale sampling revealed substantial heterogeneity in vector density and distribution between villages and districts. Vector-habitat associations predicted from this larger-scale surveillance differed from those inferred from smaller-scale studies in Kudat; highlighting the importance of local ecological context. Findings highlight potential trade-offs between maximizing temporal versus spatial breadth when designing entomological

**Funding:** This work was funded by the Biotechnology and Biological Sciences Research Council Doctoral Training Programme (Grant No. 1517720; https://bbsrc.ukri.org/skills/investing-doctoral-training/dtp/ to HF and RB) and the Medical Research Council (Grant No. G1100796; https://mrc.ukri.org/funding/browse/esei-specification/environmental-and-social-ecology-of-human-infectious-diseases-esei-specification/ to TC, KF, CD, IV and HF). The funders had no role in study design, data collection and analysis, decision to publish or preparation of the manuscript.

**Competing interests:** The authors have declared that no competing interests exist.

surveillance; and provide baseline entomological and epidemiological data to inform future studies of entomological risk factors for human *P. knowlesi* infection.

## Author summary

The primate malaria parasite, *Plasmodium knowlesi*, is a common cause of human malaria in Malaysian Borneo. Most knowledge about the ecology and behaviour of mosquitoes transmitting *P. knowlesi* in Borneo comes from a limited number of sites near the major epicentre of human infection in Kudat District, Sabah. On this basis, human exposure to vectors was predicted to be higher in human settlement areas than in farming or forest habitats. Here we aimed to characterise the diversity and abundance of *P. knowlesi* vectors over a wider area of Sabah to test hypotheses about vector-habitat relationships and associated human exposure risk. Working in 11 villages across 4 districts in Sabah, we found low densities of the *P. knowlesi* vector, *An. balabacensis*. However, vector densities were higher in farm and forest habitats than in villages across this broader area, in contrast to findings from small scale study in Kudat. No association was observed between mean *An. balabacensis* abundance and *P. knowlesi* seropositivity in communities; however the ability to detect such an association, even if present, was limited by the relatively small number of mosquitoes collected.

## Introduction

Human infection with the simian malaria parasite, *Plasmodium knowlesi* is now widespread across South East Asia with a large focus of transmission occurring in Malaysian Borneo. *Anopheles* mosquitoes in the Leucosphyrus complex are responsible for transmitting *P. knowlesi* [1], and the species *An. balabacensis* has been confirmed as the primary vector in the largest hotspot of human infection in the Kudat district of Sabah, Malaysian Borneo. Identification of vector species responsible for *P. knowlesi* transmission and habitats associated with human exposure is a vital first step for planning control measures. However, most of our current understanding of *P. knowlesi* ecology comes from intensive study within the Kudat epicentre [2–4]. Although human *P. knowlesi* cases have been reported throughout the state of Sabah [5], detailed study of vector ecology has been mostly restricted to a 2x3 km intensive study site in Kudat (Fig 1) and two sites on the neighbouring Banggi island [2]. One study compared *An. balabacensis* vector density, infection rates and survival in a village, plantation

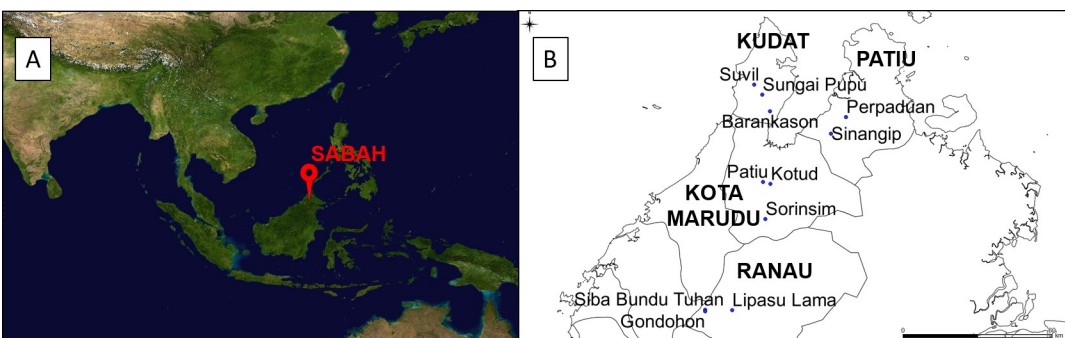

**Fig 1.** A) Location of Sabah in Malaysian Borneo (Image source: https://commons.wikimedia.org/wiki/Atlas_of_the_world) and B) Map of Northern Sabah indicating the eleven villages across 4 districts where entomological sampling was conducted in this study between March to June 2016.

and secondary forest site [2], and concluded that *An. balabacensis* density was highest in the peri-domestic setting; challenging the previous paradigm of humans only being at risk if spending long periods of time in the forest [6]. However, this study also found that vector survival and infection rates were higher at forest and farm sites than those in the village [2]; indicating a higher "per mosquito bite" risk of infection in these habitats. Further investigations in villages in Kudat indicated that *An. balabacensis* were common in peri-domestic settings, and more abundant at households where human *P. knowlesi* cases were reported than homes of uninfected controls [3]. The higher abundance of vectors in peri-domestic settings and association between peri-domestic vector abundance and human cases suggests exposure may also be happening primarily around houses. However, a more recent study at one site in Kudat reported no difference in *An. balabacensis* density at peri-domestic, farm and forest edge habitats [7]. Most recently, Fornace *et al* [8] combined data on habitat-specific vector abundance and human movement data within Kudat District to show that most human exposure likely occurs in areas close to both secondary forest and houses. While this body of work is vital for understanding the current transmission hotspot centred in Kudat [2,3,8], it remains unclear how generalizable these findings are to other areas of Sabah, Malaysia or SE Asia in general where *P. knowlesi* is emerging.

Longitudinal sampling at three sentinel sites in Kudat demonstrated that *An. balabacensis* is the dominant vector species (95.1%) [2]. However, other members of the Leucosphyrus complex and *An. donaldi* have been implicated in *P. knowlesi* transmission in other parts of Malaysia [2,3,9–14]. Evidence suggests that there is substantial heterogeneity in vector diversity and density even between villages only two kilometres apart due to environmental factors such as land-cover, type of agriculture, availability of mosquito breeding sites, temperature, topography and elevation [15–17]. The landscape in Kudat is a fragmented mix of forest, farm and deforested areas, but is relatively similar in altitude, with no major urbanization. However, across the state of Sabah, there is substantial variation in elevation, the size and distribution of forest areas, and local agricultural activities thus it is likely that *P. knowlesi* vector ecology in Kudat district may not fully represent the state as a whole.

Vector density and sporozoite infection rates are key entomological indicators frequently investigated as proxies of human exposure risk [18]. Vector density has been associated with human *Plasmodium* prevalence and incidence in some contexts [19–23], but not others [24–27]. Entomological indicators may not be robust predictors of infection burden given the non-linear relationship between entomological inoculation rates (product of vector biting and infection rates) and *Plasmodium* prevalence [28]. Reliable entomological predictors of zoonotic malaria risk for humans may be especially difficult to define due to the additional interaction between vectors and macaque reservoir populations. At present, no robust entomological predictors of *P. knowlesi* human infection risk have been defined.

Investigation of entomological indicators of *Plasmodium* infection requires high resolution, spatially and temporally concurrent data on vector bionomics and infection prevalence or incidence. A particular challenge in the study of *P. knowlesi* epidemiology is that human infection rates are generally very low, thus requiring often prohibitively large sample sizes to reliably estimate prevalence. Given these difficulties in measuring "active" infection, serology may provide a more tractable alternative for indirectly measuring previous infection. As part of an interdisciplinary study o*n P. knowlesi* epidemiology ["MonkeyBar" project, [29]], a large-scale cross-sectional survey was conducted throughout Sabah State, Malaysia, to estimate human exposure based on sero-prevalence (September to December 2015) [30]. This provided a unique opportunity to carry out complimentary entomological surveillance to assess spatial heterogeneity in *P. knowlesi* vector abundance and its concordance with human infection risk as estimated from serology.

The goal of this study was to investigate *P. knowlesi* vector species, density and infection rates across wider spatial scales in Sabah. Key aims were to identify associations with habitat type (forest, farm and village) to identify where human biting risk is highest. In addition, we investigated village-level associations between mean vector abundance and human *P. knowlesi* infection risk as estimated from the Monkeybar sero-prevalence study.

## Methods

### Village selection

A subset of 11 villages were selected from a larger group in Sabah province, Malaysia where a cross-sectional survey of *P. knowlesi* sero-positivity in people was conducted (September to December 2015 [29]). Three to four villages were selected from 4 districts to encompass a range of altitudes and habitat types: Kudat (altitude: 4–223m), Kota Marudu (8–745m), Pitas (7–218m) and Ranau (53–1275m) (Fig 1). Entomological sampling was carried out in all 11 villages approximately six months after the human sero-prevalence survey. All 11 villages were consecutively sampled over a 3-month period (21/03/16–16/06/16). One village was sampled per week, with mosquito collections being conducted over four consecutive nights. The research team attempted to visit a village from a different district on each week, so that district-level differences were not confounded by temporal autocorrelation. However this was not always logistically possible (see S1 Table for sampling dates).

### Study sites within villages

Villages were accessible by tertiary or dirt track roads. All villages were rural, with small populations of < 750 residents. These were generally structured as a group of houses surrounded by a mosaic of crops (usually largely palm oil and rubber trees) and secondary forest patches. Thus there was a range of habitats available at each village. Within each village, mosquitoes were collected in three distinct habitat types: forest patch, farm and peri-domestic settings (e.g. Fig 2). This range of habitats replicated the sampling design used in a previous study in Kudat district [2]. The peri-domestic environment was defined as the outdoor garden area immediately surrounding a household (outside, < 5m from the main house). Farm sites were located in small plantations, and forest sites were in patches of secondary forest comprising non-agricultural trees. Due to the wide geographical range of our sampling, the farm habitat varied between villages depending on what crops were locally cultivated (S1 Table). Forest was distributed patchily throughout the area with patch sizes varying significantly between villages (0.075–10km$^2$, S1 Table).

Mosquito sampling sites were selected by walking in and around each village at the start of each visit to identify all accessible locations within each of the 3 habitat types. One location per habitat type was selected based on the following criteria: peri-domestic- consent from household residents, farm- a point at least 25m from the nearest house to differentiate from peri-domestic sites, forest- a minimum patch size of 10x10m, with sampling occurring at least 20m from forest edge (if not possible, then centre of forest patch). On each night of sampling, one team of two people performed Human Landing Catches (HLC, details below) in each of the 3 habitat types, then the teams rotated between habitats on subsequent nights. Across all four sampling nights, a different sampling point was selected within each of the 3 focal habitat types each night. Each sampling point was at least 25 m from the location used the previous night. Only three nights of collections were performed for Sungai Pupu and Patiu villages due to heavy rainfall and fogging (for dengue control) taking place on the fourth day.

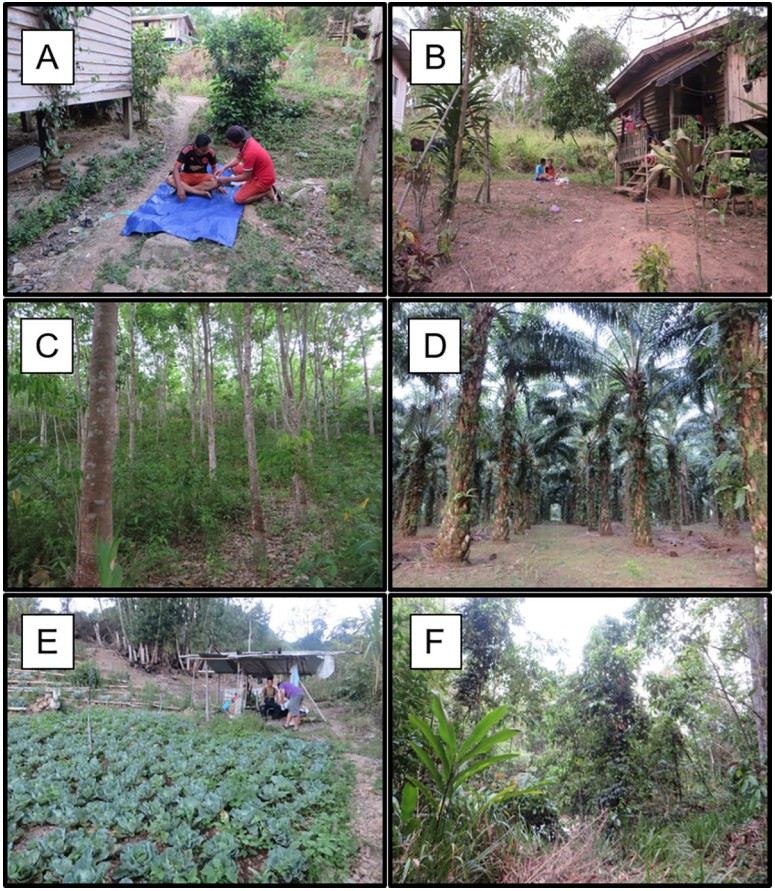

**Fig 2.** Photos showing examples of typical peri-domestic (A, B), farm (C = rubber, D = palm, E = cabbage) and forest (F) habitats where mosquito collections were conducted in this study.

## Mosquito sampling

Mosquitoes were collected using the Human Landing Catch (HLC) technique [2]. Previous studies in Sabah have evaluated a range of different trapping methods for *P. knowlesi* vectors (e.g CDC light traps, e-nets, monkey-baited traps, resting traps [31,32]) and found HLC to be the most effective. Briefly, volunteers were positioned in teams of two with their lower legs exposed, and trapped mosquitoes which landed on them to feed using 30ml plastic screw-top vials. One mosquito was trapped per vial and the hour and habitat of collection were recorded on each. Nightly collections per site represented the total number of mosquitoes caught in HLC carried out by two people. Catches were performed between 18:00–00:00 to include the peak biting time of Sabah's primary *P. knowlesi* vector, *An. balabacensis* [2,33]. Previous studies in this area have shown the majority of *An. balabacensis* biting activity occurs before midnight [3,8]. All HLCs were conducted outdoors because *P. knowlesi* vectors exhibit exophilic host seeking behaviour [12].

## Mosquito processing

At the end of each 6-hour sampling period, mosquitoes trapped inside vials were transported to the central field station and put in a -20˚C freezer. Mosquitoes were killed by storing at -20˚C overnight and identified to genera and species level the following day using the keys of

Rattanarithikul *et al* (2005/6) [34–37]. Species belonging to the Leucosphyrus group were identified using [38]. Specimens were stored in 95% ethanol until further processing.

### *Plasmodium* detection in *Anopheles*

DNA was prepared from all Leucosphyrus group *Anopheles*, and *An. donaldi* and *An. maculatus* (known malaria vectors, [14,33,39]). First the ethanol preservative was removed from these sample tubes and then DNA extracted from the whole body using the QIAGEN DNeasy Blood and Tissue Kit following the manufacturer's instructions with the following minor modifications. Specimens were initially ground in 180 µl buffer ATL using a pestle and hand-held homogenisor, and lastly eluted in a volume of 25 µl TE buffer. Nested PCRs were conducted to screen samples for *Plasmodium* DNA using the method of Snounou and Singh [40], which identifies DNA of any species within the *Plasmodium* genus. Samples positive for *Plasmodium* DNA were subjected to a further PCR to identify the species present. Nine separate reactions were set up following the method of Ta *et al* [41] (to detect *P. falciparum*, *P. vivax*, *P. malariae* and *P. ovale*), Lee *et al* [42] (*P. coatneyi*, *P. inui* and *P. cynomolgi*) and Imwong *et al* [43] (*P. knowlesi*) (S2 Table).

### *Plasmodium knowlesi* sero-prevalence in humans

Sero-prevalence data on *Plasmodium* exposure in humans in the study villages was obtained from a cross-sectional survey as described in [30]. In summary, no active *Plasmodium* infections were observed by either microscopy or PCR [29] in this survey, thus serological measures of previous *P. knowlesi* exposure were used to examine associations with the density of Leucosphyrus group *Anopheles* [44]. Measures of village level sero-positivity (the proportion of individuals from the total screened per village that were IgG positive for *P. knowlesi*) were estimated for the 11 villages in which entomological surveillance was conducted [29]. Serological screening can detect individuals infected with *P. knowlesi* at least within the previous 28 days [44] but it is unknown how long these antibodies can persist for.

### Data analysis

**Anopheles diversity across habitat types.** Data were analysed using the R statistical programming software, version 3.4.2. The "vegan" package was used to measure four species diversity indices: species richness, rarefied species richness, Simpson's index and the Shannon index. These measures were used to estimate and compare *Anopheles* diversity across habitat types (peri-domestic area, farm and forest). Species richness is the total number of different *Anopheles* species collected in each village. The rarefied species richness is the species richness if collections had the same *Anopheles* density (ie. set to the group with the lowest total density). Rarefaction is a method used to standardise unequal sampling sizes [45,46]. The Simpson's index,

$[\lambda = (n/n\text{-}1) \text{ x } \sum p_s (1\text{-}p_s)]$ [47], where
n = total *Anopheles* density
$p_s$ = each species count/n,

measures the probability that two individuals randomly sampled from the dataset will be of the same species [48]. The Simpson's Index is noted to be sensitive to abundant species [49], thus the Shannon Index was also calculated as a comparison. The Shannon index,

$H = \text{-}\sum (n/N) \log (n/N)$ (48), where
N = total *Anopheles* density
$n_i$ = each species count,

measures the uncertainty in predicting the species of an individual randomly sampled from the dataset [49]. Confidence intervals for Simpson's Diversity Index were calculated following Zhang [47].

**Analysis of environmental variables.**   Percentage forest cover in a 100m buffer (circle of radius 100m) around each sampling location for HLC was calculated using the Hansen global forest cover 2014 map, with forest defined as 50% canopy cover [50]. GLMMs were constructed in R using the lme4 package to extract the mean elevations and proportion of forest cover at all mosquito collection sites. A negative binomial model was used to predict mean elevation and a model with a binomial distribution was used for percentage forest cover. Elevation and percentage forest cover were the response variables and habitat was the explanatory variable, with date and village set as random effects.

**Mosquito presence and density analyses.**   Statistical analysis was performed on two sets of mosquito data: 1) *An. balabacensis* only, and 2) All Leucosphyrus group *Anopheles*. The second group was inclusive of *An. balabacensis* (n = 32), *An. latens* (n = 7) and suspected *An. balabacensis/An. latens* (n = 2); defined as being either of these two species with identification to species level not being possible due to the loss of fragile scales on the wings necessary for morphological identification. Both *An. balabacensis* or *An. latens* are implicated in the transmission of *P. knowlesi* in Malaysian Borneo [2,9] thus were analysed as a whole. The packages lme4 and multcomp were used to analyse mosquito presence and density. Generalised Linear Mixed Models (GLMMs) were constructed to test for associations between the two response variables of mosquito presence (binary outcome, 0 = absent, 1 = present) and density (mean number caught per site per night), and the following explanatory variables: elevation, habitat type and forest cover. To relieve issues with scaling, elevation was converted from a continuous to a categorical variable by splitting into three elevation ranges: low (0 – 375m), medium (376 – 750m) and high (751 – 1125m). Models were fit with a negative binomial distribution for mosquito density and a binomial distribution for mosquito presence. In all models, random effects were included for village and date. The significance of explanatory variables in each of the models was tested by backward elimination using likelihood ratio tests. A Tukeys' post hoc test was performed to assess differences between each of the 3 habitat types.

**Biting time in *Plasmodium* vector species.**   The lme4 package was used to construct GLMMs in R to extract hourly biting rates of different *Plasmodium* vector species caught. Only *An. balabacensis*, *An. donaldi* and *An. maculatus* were examined because the overall density of *An. latens* (n = 7) was too low to analyse in this way. The number of mosquitoes of each species caught per hour throughout the night was examined, with the first hour as 18:00–19:00 and the last as 23:00–00:00. Hourly mosquito abundance was treated as the response variable with the main fixed effect being biting hour. A negative binomial distribution was used with date and village set as random effects. A Tukey's post-hoc test (package multcomp) was used to assess differences in biting rates between hours within each species.

**Associations between vector density and human *P. knowlesi* exposure.**   General linear models (GLMs) were constructed to test for associations between mosquito presence and density for 1) *An. balabacensis* only and 2) Leucosphyrus group *Anopheles* (*An. balabacensis/An. latens*) and village-level *P. knowlesi* sero-positivity. A GLMM with a negative binomial distribution was used to predict mean mosquito density from each village where mosquito density per night was the response variable and habitat and date were fit as random effects. A binomial GLMM was used to predict the probability of detecting a mosquito in each village where mosquito presence (1) or absence (0) per night was the response variable and habitat and date were fit as random effects. These village-specific estimates of mean vector density and probability of occurrence were used to test for associations with the proportion of individuals sero-positive for *P. knowlesi* antigens in each village. A binomial GLM was used with village sero-

positivity as the response variable and mosquito presence or density as the fixed effect. Entomological collections began ~ 6 months after the cross-sectional survey thus did not run in parallel with human sampling. However, an assumption of this analysis is that entomological measures were assumed to be reflective of general differences between villages at the time of the cross-sectional survey. A post hoc power analysis was performed using the pwr.f2.test function from the pwr package in R (effect size = $R^2/(1–R^2)$), significance level = 0.05, power = 0.8) to determine the sample size required to detect an association between village level human *P. knowlesi* sero-positivity rates and *P. knowlesi* vector presence or abundance. All data collected in this study is available from Harvard dataverse "https://doi.org/10.7910/DVN/3QG1HP".

## Ethics statement

This project was approved by the Malaysian Ministry of Health (NMRR-12-786-13048) and by the research ethics committees of the London School of Hygiene and Tropical Medicine (Ref. 6302). Homeowners gave permission to use the area around their houses for mosquito collection. All volunteers who carried out mosquito collections were adults and signed informed consent forms prior to the study. Volunteers were provided with antimalarial prophylaxis during participation and one month after performing HLC, volunteers were screened for *Plasmodium* by giemsa stained thick and thin blood smears. Participants were asked to immediately report if they felt ill or feverish and would be taken to the nearest medical facility for check-up and malaria treatment if required. No participants reported malaria infections during the study.

## Results

In 42 nights of sampling, a total of 5588 mosquitoes belonging to eight genera were collected (S3 Table). The majority of specimens were from the *Culex* and *Armigeres* genera, with only 4% *Anopheles* and 8% *Aedes*. Five genera were found in peri-domestic habitats, six in farm and seven in forests (S1 Fig). Species known to transmit *Plasmodium* in Sabah (*Anopheles balabacensis*, *An. latens* and *An. donaldi)* comprised 1.1% of the total mosquito catch. Six species of *Anopheles* were collected (Table 1) with *An. maculatus* and *An. barbumbrosus* being the most abundant. Dengue vector species (*Ae. albopictus* and *Ae. aegypti*) comprised 6.9% of mosquitoes collected. The majority of *Aedes* specimens were *Ae. albopictus* (~90%) with only a few *Ae. aegypti* (~1%). The remaining *Aedes* specimens could not be identified to species level.

Anopheline species diversity was lower in peri-domestic and farm sites than at forest sites (Table 2). Both the rarefied species richness, Shannon and Simpson Indices estimated similar trends with forest sites having higher *Anopheles* species diversity, followed by farm sites and then peri-domestic sites (Table 2).

Averaging over all villages, there was no systematic difference in the mean altitude of the forest, farm and peri-domestic mosquito sampling sites ($P > 0.05$, S4 Table); thus habitat type was not confounded by altitudinal variation. As expected, the percentage of tree cover above sampling points in farms and forests was higher than in peri-domestic settings, however this result was not significant (($P > 0.05$, S4 Table).

### Vector density and distribution

The major *P. knowlesi* vector *An. balabacensis* (n = 32) was found in approximately 14% of collections, with no significant difference in probability of detection between habitats ($X^2 = 5.33$, $df = 2$, $P = 0.07$), or in association with percentage forest cover ($X^2 = 3.16$, $df = 1$, $P = 0.08$) or elevation ($X^2 = 0.21$, $df = 2$, $P = 0.90$). Pooling all *Anopheles* species in the Leucosphyrus group (n = 41), the probability of detection varied with habitat ($X^2 = 7.42$, $df = 2$, $P = 0.02$) but not

**Table 1. *Anopheles* species caught in eleven villages within the four districts: Kudat, Kota Marudu, Pitas and Ranau in Sabah, sampled from March to June 2016.** Village names: SUV–Suvil, SUN–Sungai Pupu, BAR–Barankason, SOR–Sorinsim, PAT–Patiu, KOT–Kotud, PER–Perpaduan, SIN–Sinangip, LIP–Lipasu Lama, SIB–Siba Bundu Tuhan and GON—Gondohon.

| Mosquito genera/ species | District of sampling | | | | | | | | | | | |
| --- | --- | --- | --- | --- | --- | --- | --- | --- | --- | --- | --- | --- |
| | Kudat (villages) | | | Kota Marudu (villages) | | | Pitas (villages) | | Ranau (villages) | | | |
| | SUV | SUN | BAR | SOR | PAT | KOT | PER | SIN | LIP | SIB | GON | Total (%) |
| Leucosphyrus gp. | 0 | 1 | 0 | 8 | 1 | 1 | 0 | 10 | 19 | 0 | 0 | 41 (19.3) |
| *An. balabacensis*[*][#] | 0 | 1 | 0 | 3 | 1 | 1 | 0 | 7 | 12 | 0 | 0 | 32 (15.1) |
| *An. latens*[*] | 0 | 0 | 0 | 0 | 0 | 0 | 0 | 0 | 7 | 0 | 0 | 7 (3.3) |
| *An. balabcensis or An. latens*[*] | 0 | 0 | 0 | 1 | 0 | 0 | 0 | 1 | 0 | 0 | 0 | 2 (0.9) |
| Barbirostris *gp* | 3 | 33 | 1 | 19 | 3 | 3 | 0 | 23 | 0 | 1 | 3 | 89 (42.0) |
| *An. barbumbrosus* | 0 | 16 | 1 | 14 | 3 | 3 | 0 | 22 | 0 | 0 | 2 | 61 (28.8) |
| *An. donaldi*[#] | 3 | 16 | 0 | 2 | 0 | 0 | 0 | 0 | 0 | 1 | 1 | 23 (10.9) |
| *An. maculatus*[#] | 0 | 0 | 0 | 8 | 3 | 29 | 0 | 10 | 0 | 29 | 1 | 80 (37.7) |
| *An. tesselatus* | 0 | 0 | 0 | 0 | 0 | 0 | 2 | 0 | 0 | 0 | 0 | 2 (0.9) |
| Total *Anopheles* sp. | 3 | 34 | 1 | 34 | 8 | 33 | 2 | 43 | 19 | 31 | 4 | 212 |

[*] vector of *P. knowlesi*

[#] vector of *P. falciparum/ P. vivax*

forest cover ($X^2$ = 3.31, $df$ = 1, $P$ = 0.07) or elevation ($X^2$ = 0.34, $df$ = 2, $P$ = 0.85). Leucosphyrus group mosquitoes were more likely to be caught in farm ($P$ = 0.02) and forest ($P$ = 0.02) sites than in peri-domestic environments (Fig 3A).

The density of *An. balabacensis* varied with habitat ($X^2$ = 9.82, $df$ = 2, $P$ < 0.01) but not with elevation ($X^2$ = 0.13, $df$ = 2, $P$ = 0.93) or percentage forest cover ($X^2$ = 3.16, $df$ = 1, $P$ = 0.08). *Anopheles balabacensis* was significantly more abundant in farm ($P$ < 0.01) and forest ($P$ < 0.01) habitats than in peri-domestic areas (Fig 3B, S2 Fig for raw data). Similarly, habitat was a significant predictor of the mean density of the Leucosphyrus group in general ($X^2$ = 12.92, $df$ = 2, $P$ < 0.01); with their density being significantly lower in peri-domestic environments than in farm ($P$ < 0.001) or forest ($P$ < 0.001) habitats (Fig 3C, S2 Fig for raw data). The mean density of the Leucosphyrus group did not vary in relation to local forest cover ($X^2$ = 4.12, $df$ = 1, $P$ = 0.04) or elevation of the collection site after accounting for habitat differences ($X^2$ = 1.64, $df$ = 1, $P$ = 0.20).

## Biting patterns of *Plasmodium* vector species

The biting patterns of the three known *Plasmodium* vector species (*An. maculatus*, *An. donaldi* and *An. balabacensis)* were assessed. Too few *Anopheles latens* individuals (a vector of *P. knowlesi*) were sampled for robust description (n = 7). These three species were found during all sampling hours (18:00–00:00), with some tendency for higher activity during the early evening hours (18:00–20:00). However due to the relatively small numbers collected and high variability in hourly catches, no clear peaks in biting time were evident (Tukey's, $P$ > 0.05, S3 Fig).

**Table 2. Measures of diversity in *Anopheles* species across different habitat types sampled in eleven villages in Sabah from March to June 2016.**

| Habitat | *Anopheles* abundance | Species richness | Rarefied species richness | Shannon index | Simpson's index | Simpson's index ± 95% confidence intervals |
| --- | --- | --- | --- | --- | --- | --- |
| Peri-domestic | 22 | 4 | 2.380 | 0.969 | 0.5 | 0.52 ± 0.22 |
| Farm | 85 | 4 | 2.858 | 1.276 | 0.694 | 0.73 ± 0.05 |
| Forest | 98 | 5 | 3.139 | 1.477 | 0.750 | 0.79 ± 0.04 |

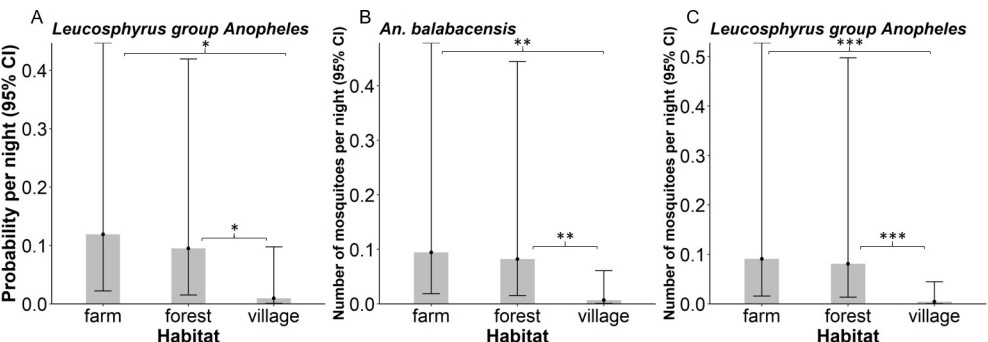

**Fig 3.** Predicted A) Probability of catching Leucosphyrus group *Anopheles* B) Mean density of *An. balabacensis* and C) Mean density of Leucosphyrus group *Anopheles* in farm, forest and peri-domestic habitats. Error bars represent 95% confidence intervals.

### *Plasmodium* infection rates

Of the 144 female mosquitoes that were potential *Plasmodium* vector species (*An.* Leucosphyrus gp, *An. donaldi* and *An. maculatus*), only one tested positive for *Plasmodium*. This was an *An. balabacensis* collected in a forest patch in Sinangip village, Pitas, which was infected with *P. knowlesi*. This represents an infection rate of ~3% (n = 1/32) in *An. balabacensis*.

### Association between *Plasmodium* vector density and human *P. knowlesi* exposure

Seroprevalence rates of *P. knowlesi* in people across the study area were provided by the Monkeybar large cross-sectional survey [8]. Within the subset of 11 villages where entomological surveillance was conducted, sero-positivity rates ranged from 0% (Sib and Sun) to 13.9% (in Sor). In these villages, the probability of trapping *An. balabacensis* per night in an HLC ranged from 0.11–0.42 (Fig 4A), and 0.11–0.50 for the Leucosphyrus group overall (Fig 4B). No significant relationship was detected between human *P. knowlesi* sero-positivity rates and the probability of detecting *An. balabacensis* or Leucosphyrus group *Anopheles* ($P > 0.05$, S5 Table). No significant relationship was detected between human *P. knowlesi* sero-positivity rates and the density of *An. balabacensis* as measured 6 months afterwards (Fig 4C) or Leucosphyrus group *Anopheles* (Fig 4D) ($P > 0.05$, S5 Table). However, post hoc power analysis indicated that the study had limited power to detect an association between seropositivity rates and vector abundance given the lower than anticipated densities of primary vector species. With the low vector densities observed here, and based on the GLM described above, 142 and 390 villages respectively would need to be sampled to detect a positive significant relationship ($P = 0.05$) between the presence of *An. balabacensis* and *An. Leucosphyrus* gp. and human *P. knowlesi* seropositivity rates with 80% power (S5 Table). The sample sizes required to detect a positive significant relationship ($P = 0.05$) between the density of *An. balabacensis* and *An. Leucosphyrus* gp. and human *P. knowlesi* seropositivity rates with 80% power would be derived from sampling 159 and 113 villages respectively (S5 Table).

## Discussion

Here we describe the density and diversity of *P. knowlesi* vectors across 4 districts in Malaysian Borneo where this parasite is a significant public health problem. This entomological surveillance covered a wider geographical region than has been investigated before. We found that *An. balabacensis*, the *P. knowlesi* vector, is widely distributed across 4 districts of Sabah,

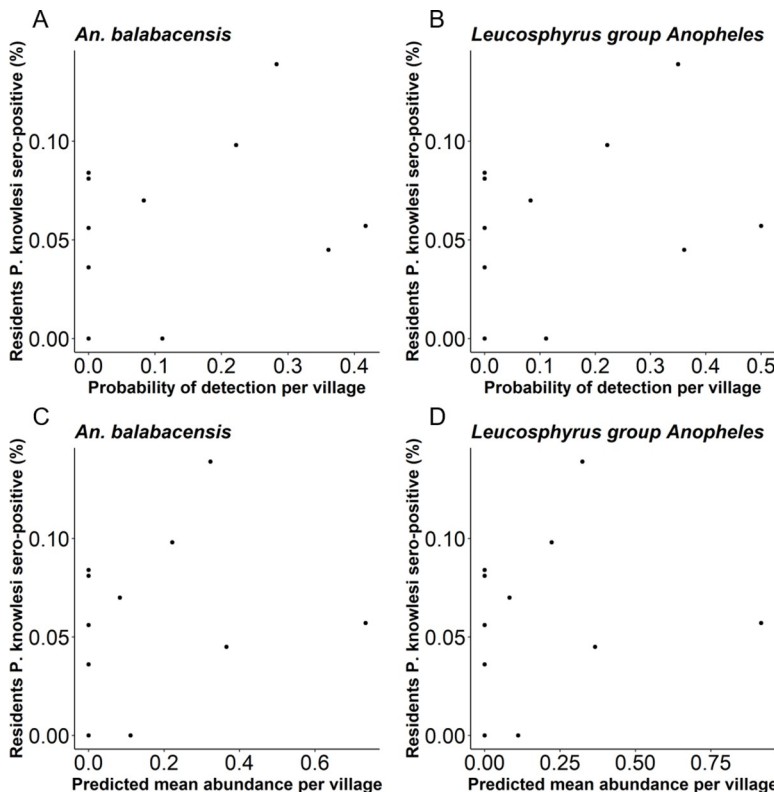

**Fig 4.** Association between the proportion of individuals in a village sero-positive for *P. knowlesi* antigens and the A) detection of *An. balabacensis* B) detection of Leucosphyrus group *Anopheles* C) density of *An. balabacensis* and D) density of Leucosphyrus group *Anopheles* caught in the village per night.

Malaysian Borneo; but at a lower relative abundance (within the Anopheline community) than has been previously reported near the epicentre of human cases in Kudat. There was substantial heterogeneity in the density and diversity of vector populations both within and between districts. Vector surveillance over this wider geographic area indicated a different pattern of vector-habitat relationships than hypothesized from single site studies in Kudat; with vector abundance being higher in forest and farm habitats than in peri-domestic environments. Using human *P. knowlesi* sero-positivity data gathered from a large cross-sectional survey, no significant correlation was detected between village-level human infection exposure and vector density. However, power to detect such an association was limited by low vector density throughout the study area and resultant small sample sizes. Larger-scale and longer-term studies thus may be required for robust investigation of entomological indicators of *P. knowlesi* infection.

*Anopheles balabacensis*, the primary vector incriminated in *P. knowlesi* transmission in Sabah, was found throughout the study area but at considerably lower density than previously estimated in focal studies around Kudat. Based on surveillance at a few sites in Kudat, this vector was previously reported to be the dominant Anopheline biting humans (e.g. 95.1% of Anophelines, [2,3]). However, *An. balabacensis* accounted for only 15.1% of the human-biting Anophelines in this study. We note that during the study there were droughts across Sabah due to the El Nino; which could have had impact on vector densities. However it is difficult to assess the potential effects on vectors from available data (collated in [8]) because there were no sites consistently sampled before, during and after the El Nino. Another factor that could

account for the lower estimates of *An. balabacensis* density observed in this study compared to previous ones in Kudat is the duration and selection of sampling sites. In a previous study in Kudat, the farm, forest and village site were selected based on having high *An. balabacensis* density, as required to generate sufficient sample sizes for parasite screening [2]. Here mosquitoes were sampled for only 3–4 nights per site over a 3-month period (in 2016), whereas previous work sampled mosquitoes over 12 months (3 nights/month, 2013–2014). If vector population dynamics are highly seasonal, this shorter-term sampling could substantially over or underestimate average annual densities. However, previous studies indicate there is little seasonality in *An. balabacensis* abundance with vector numbers staying relatively constant across months in this area [2,30]. Given the potentially minor role of seasonality in these vector populations, the shorter period of sampling used here may be sufficient to reflect general differences in vector abundance between villages and sites. However, a more detailed understanding of temporal variation in these vector populations would be of great value for refining estimates of spatial variation.

Habitat type was a major predictor of *Anopheles* presence and density in this study. Both *An. balabacensis* and the Leucosphyrus group were more abundant in farm and forest than peri-domestic habitats. Similar vector-habitat associations have previously been reported in Kapit, Sarawak [9], and in Peninsular Malaysia [12], but a previous study conducted in a single farm, forest and peri-domestic site in Kudat found *An. balabacensis* to be most abundant in the village [2]. A further study conducted in Kudat found similar densities of *An. balabacensis* at all habitat types sampled (peri-domestic, farm and forest edge) [7]. Differences reported in Wong *et al* [2] and in Chua *et al* [7] may have been due to site specific factors rather than habitat, highlighting the need for replicated sampling over wide geographical areas for robust habitat prediction [51]. The sampling period applied here (11 villages, 3–4 nights per village) was shorter than the longer period used previously in Kudat (3 sites, 2–3 nights per month for 12 months) [2]. These differences in sampling design limit direct comparison of vector densities between these studies, however it does provide an opportunity to make qualitative comparisons of vector density between sites, even if more precise quantification is limited by the shorter collection period. Future studies investigating vector habitat associations over wider geographic regions in Sabah would require substantial depth (time and resources) to rigorously assess differences with previous studies conducted in the Kudat district.

Recent epidemiological studies have identified forest and agricultural-related work activities as risk factors for *P. knowlesi* infection in Sabah [29,52]. Forest cover and historical forest loss have also been significantly associated with the occurrence of human cases of *P. knowlesi* in this area [53] and that increasing distance from the forest reduces the chance of being bitten by an infected mosquito [8]. Additionally, investigations into human movement patterns in rural villages in Sabah indicate that during mosquito biting hours (18:00–06:00), people are less likely to use areas further away from the home indicating that people are at highest risk of exposure if their houses are in closer proximity to forested areas [8]. Therefore whilst our study found highest densities of *P. knowlesi* vectors in forest and farm sites, people are less likely to use these areas throughout the night, and so it is the proximity of the home to these habitats that is the key risk for human exposure.

Only one *Plasmodium* infected mosquito was found across the study area, an *An. balabacensis* infected with *P. knowlesi* caught in a forest patch, corresponding to a total infection rate of ~3% (1 out of 32 tested). Whilst this is in line with the expectation that *P. knowlesi* infection rates are highest in *An. balabacensis* found in forests [2], the sample size of infected mosquitoes was too low to draw any significant conclusions about habitat-dependent mosquito infection rates. In previous studies in Kudat, the *P. knowlesi* infection rate of *An. balabacensis* ranged from 0–0.88% [2–4,7], with overall infection rates (all *Plasmodium* species) ranging from

1.45–3% [2–4,7]. No *P. knowlesi* infections were detected in *An. balabacensis* caught in Kudat here, which may be due to the low number of samples tested. However, we note that the absence of *P. knowlesi* in the *An. balabacensis* tested from Kudat here coincides with a recent reduction in the number of human *P. knowlesi* cases in this district [54]. The sample size of *An. balabacensis* obtained in this study was too low to draw conclusions on infection rates across different habitats or districts. Given the low vector densities and sporozoite rates in the study area, quantification of spatial variation in mosquito infection rates would likely require years of continual sampling in a large number of sites. Such high-resolution intensive sampling was not possible within the scope of this project. This highlights the trade-offs between spatial breadth and temporal depth that must be considered in designing surveillance programmes for zoonotic vectors. Given the difficulty of achieving both depth and breadth, the best solution may be a mixed approach combining short-term sampling at a wide range of sites coupled with intensive long-term sampling at a smaller number of fixed sentinel sites. A further factor influencing vector infection (and density) could be the presence and abundance of the macaque reservoir in the study area. Collecting this type of data is labour intensive and was outwith the scope of this study however is an important part of this malaria system that future studies should consider.

Altitude has long been recognized as a significant predictor of malaria transmission [55–60], however it was not a significant predictor of *Anopheles* presence and density across the wide gradient investigated here (13–1125). Across villages investigated here, there is substantial variation in climate and local tree species as well as elevation thus the significant additional environmental heterogeneity introduced by sampling over such a wide geographical range which may have swamped the more modest impact of elevation. All sites sampled may also have been within the altitudinal/temperature range suitable for *An. balabacensis*.

No significant association between the density of mosquito vectors (*An. balabacensis* and all *An.* Leucosphyrus group mosquitoes) and human sero-positivity for *P. knowlesi* at the village-level was detected. A notable limitation of our study design was that entomological sampling was performed six months after human data was collected. Thus, there was a temporal mismatch in the timing of human and entomological sampling which could have limited the strength of any association. The entomological data was only collected for a few days and may not have been an accurate representation of vector conditions at the time of human sampling. Alternatively, even if sampling was conducted concurrently there may be no link between vector density and human infection. Malaria vector densities do not always correlate with human risk [24–27]; with a lack of synchrony perhaps being more likely with zoonotic malaria due to the additional complexity introduced by the macaque reservoir. However, our ability to test these hypotheses was limited by a lack of statistical power; which post hoc analysis indicated that considerably larger sample sizes (approximately x 14) would have been required to robustly detect an association between *An. balabacensis* vector density and human sero-positivity with (80%) power. Thus although data collected here was not sufficient for robust analysis of entomological indicators, it can provide a useful guide for the design of future studies into the epidemiology of this complex malaria system.

## Supporting information

**S1 Table. Villages selected for entomological sampling in Sabah from March to June 2016.** Characteristics of the eleven villages in Sabah State, Malaysia, where mosquito vectors were sampled. "Crops" describes the dominant types of subsistence farming occurring in the village. "Approximate area of forest patch" refers to the size of the forest patch (estimated from map) in which mosquito collections were conducted within the forest habitat type.

"Population size" refers to the estimated number of residents derived from household enumeration conducted as part of the Monkeybar cross-sectional survey in September to December 2015.
(DOCX)

**S2 Table. Primer pairs used in nested PCR to detect parasites from *Plasmodium* genus and specific human and simian malaria species.**
(DOCX)

**S3 Table. Relative frequencies of eight mosquito genera caught in eleven villages within the four districts: Kudat, Kota Marudu, Pitas and Ranau in Sabah, sampled from March to June 2016.**
(DOCX)

**S4 Table. Mean values of elevation and percent forest cover within each of the three habitat classes where mosquito sampling was conducted in this study.**
(DOCX)

**S5 Table. Results from binomial GLMs testing for associations between vector presence and density per night and the proportion of individuals in a village sero-positive for *P. knowlesi* antigens.** GLMs were used to compute ($R^2$ and effect sizes) for subsequent power analyses to determine sample sizes required to generate the same associations ($P = 0.05$) with 80% power.
(DOCX)

**S1 Fig. Proportional representation of different mosquito genera within collections made in peri-domestic, farm and forest habitats across 11 villages in this study.**
(DOCX)

**S2 Fig. *Anopheles balabacensis* and An. Leucosphyrus group mosquitoes caught from March to June 2016 in peri-domestic, farm and forest habitats in eleven villages within four districts; Kudat, Kota Marudu, Pitas and Ranau of Sabah.**
(DOCX)

**S3 Fig. Predicted mean number of A) *An. balabacensis*, B) *An. donaldi* and C) *An. maculatus* biting per hour between 18:00–24:00 hrs, pooled across all sites and habitat types. Error bars are 95% confidence intervals.**
(DOCX)

## Acknowledgments

The authors would like to thank Albert M. Lim, Benny O. Manin and the MonkeyBar field team in Sabah for their support throughout the study, particularly field assistants Mohd Fazreen Abdullah and Nemran Bayan for their hard work. We thank the village leaders and all communities participating in entomological collections for their cooperation and interest during this research. We thank Universiti Malaysia Sabah for the use of their lab facilities.

## Author Contributions

**Formal analysis:** Rebecca Brown.

**Investigation:** Rebecca Brown.

**Methodology:** Rebecca Brown.

**Supervision:** Tock H. Chua, Heather M. Ferguson.

**Writing – original draft:** Rebecca Brown, Heather M. Ferguson.

**Writing – review & editing:** Tock H. Chua, Kimberly Fornace, Chris Drakeley, Indra Vythilingam.

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
