## [Decision Letter · Decision Letter 0]

5 Mar 2020

Dear Dr Brown,

Thank you very much for submitting your manuscript "Human exposure to zoonotic malaria vectors in village, farm and forest habitats in Sabah, Malaysian Borneo" for consideration at PLOS Neglected Tropical Diseases. As with all papers reviewed by the journal, your manuscript was reviewed by members of the editorial board and by several independent reviewers. In light of the reviews (below this email), we would like to invite the resubmission of a significantly-revised version that takes into account the reviewers' comments. 

We cannot make any decision about publication until we have seen the revised manuscript and your response to the reviewers' comments. Your revised manuscript is also likely to be sent to reviewers for further evaluation.

Sincerely,

Hans-Peter Fuehrer

Deputy Editor

Reviewer's Responses to Questions

**Key Review Criteria Required for Acceptance?**

**Methods**

-Are the objectives of the study clearly articulated with a clear testable hypothesis stated?

-Is the study design appropriate to address the stated objectives?

-Is the population clearly described and appropriate for the hypothesis being tested?

-Is the sample size sufficient to ensure adequate power to address the hypothesis being tested?

-Were correct statistical analysis used to support conclusions?

-Are there concerns about ethical or regulatory requirements being met?

Reviewer #1: Research question: Investigate environmental determinants of Plasmodium knowlesi vector density and infections rates across a wider spatial scale in Sabah. 

Key aims were to identify associations with habitat type and test whether results from small-scale sampling in one district (Kudat) are generalizable across the state. In addition, this study tested for associations between entomological variables and human exposure to P. knowlesi as measured in a related sero-prevalence study. The authors say they performed and intensive study to address their research questions. 

Methods

A larger area of Sabah was included in the study described (fig 1). Three different habitats (peri-domestic, farm and forest) per village were sampled over 4 nights. 

All sites were studied in three months (March to June 2016) the single time frame did not seem to particularly coincide with annual zoonotic malaria prevalence in Sabah. Even-so to attempt sampling 3 habitats, with teams of two per habitat, using only human landing catches (HLC) in 11 villages over a 3-month period would be expected to be limiting.

Sampling was conducted between 6pm and midnight when 6pm to 6am would have been more comprehensive.

The study design would not be expected to produce the information required to address the main research questions.

Reviewer #2: If there are no major new analyses/experiments required prior to publication.

If possible, some kind of power analysis based on detection rates here to determine the length and coverage of sampling that might be required to conclusively evaluate if vector density can be used as a proxy for human infection risk would be useful particularly for the intended audience of PLoS NTDs.

Reviewer #3: (No Response)

**Results**

-Does the analysis presented match the analysis plan?

-Are the results clearly and completely presented?

-Are the figures (Tables, Images) of sufficient quality for clarity?

Reviewer #1: While data were analysed using formulae designed to detect vector prevalence the number of vectors collected at each habitat per site were small. Measures of diversity indicated that farming areas and the forest have a greater abundance of Anopheline vectors than the peri-domestic habitats.

Reviewer #2: The analyses match the analyses plan. 

Throughout make sure that appropriate statistical test information is reported. For example, on line 369-398. What was the outcome of the GLMM that allows you to state that there is no systemic variation in altitude across site types. On line 433-434, what are the stats that show no significant effect of time on mean predicted number of bites. 

It was not clear to me why the outcome variable for vector density was the mean number caught per site per night. If date and site were both included as explanatory variables wouldn't it allow you to directly model number of mosquitoes caught? 

Tables and Figures:

Figure 1: I would remove a and include a figure that illustrates the "substantial variation in elevation, the size, and distribution of forest areas, and local agricultural activities". Could you use the Hansen global forest cover 2014 map for this?

Fig 3: I would recommend a box plot for this.

Fig 4: I don't think this figure is needed as long as the lack of statistical significance is fully reported in the text.

Reviewer #3: (No Response)

**Conclusions**

-Are the conclusions supported by the data presented?

-Are the limitations of analysis clearly described?

-Do the authors discuss how these data can be helpful to advance our understanding of the topic under study?

-Is public health relevance addressed?

Reviewer #1: At best all that can be concluded is that the sampling methods used over short time intervals may not have been sufficient to properly capture the information required to calculate the environmental determinants of vector density and infection rate at the study sites during a single short time interval.

The authors do acknowledge this but yet make statements such as An. balabacensis occurred at the study sites at lower density when compared to a much more intense and prolonged study in the Kudat District of Sabah. Had the two studies been similarly rigorous then indeed a comparison could have been made. 

In the absence of rigorous longitudinal sampling the objectives of this study on the environmental determinants of malaria vector density, particularly vectors of zoonotic malaria, would be difficult to test.

Reviewer #2: The authors identify the two key limitations of the study: the low numbers of malaria vectors sampled and the temporal mismatch between the human prevalence data and the vector density data. I thought they did a respectable job of pointing out these issues, but there are a few key points to be considered and pieces of information that I think the reader needs to assess the validity of the conclusions. 

1. Are there alternative explanations for the numbers of Anopheles in HLCs? Certainly, one explanation is that there are not many Anopheles. However, only one sampling technique was reported and it maybe that this technique was not appropriate for this particular setting. For example barrier screens have been used in other settings (Pollard et al. 2019, Parasites and Vectors) to collect blood fed Anopheles. Can you offer some kind of assessment of the probability that HLCs were not effective for Anopheles in this environment. Are there any larval surveys or alternative sampling that can support the HLC result? If HLCs are a valid technique (which in many instance they are) then what kind of power would you require to detect an association with serology data. I think this is important for recommendations moving forward. Can you use this data to determine the temporal and geographic spread you might need?

2. What was the impact of El Nino during this period. I can see from the attached Fornace publication that there was drought during the serology study which may have impacted vector densities. Was this also having an effect in 2016? Might this explain the discrepancy between site types in the two studies? During drought the oviposition may have been more likely to occur in peri-domestic habitats where people were storing water? 

3. It was not clear to me exactly what the serology data indicates and this seems imperative for helping the reader assess the association between the two data sets. From what I understood serology can be used to detect current and historical infections. On line 554 it is stated that the antigens for P. knowlesi are relatively short lived, but when I looked into the citations provided, I still could not determine how short lived. The vector data is collected in 2016 and the serology in 2015. Over what period is prevalence being captured by these data? Is it anyone exposed within 6 months, a year? We expect vector abundance to fluctuate over time at relatively short time scales compared to host infection rates and it is not clear to me why would expect human prevalence rate from a previous year (potentially a previous year with drastically different environmental factors due to el nino and calculated over a long period of time) to correlate with a snapshot of vector density taken much later and in a different season. I think the authors hit on these issues in the discussion, but they need to be much stronger in their defense of this methodology.

Reviewer #3: (No Response)

**Editorial and Data Presentation Modifications?**

Reviewer #1: (No Response)

Reviewer #2: (No Response)

Reviewer #3: (No Response)

**Summary and General Comments**

Reviewer #1: Research question: Investigate environmental determinants of Plasmodium knowlesi vector density and infections rates across a wider spatial scale in Sabah. 

Key aims were to identify associations with habitat type and test whether results from small-scale sampling in one district (Kudat) are generalizable across the state. In addition, this study tested for associations between entomological variables and human exposure to P. knowlesi as measured in a related sero-prevalence study. The authors say they performed and intensive study to address their research questions. 

Methods

A larger area of Sabah was included in the study described (fig 1). Three different habitats (peri-domestic, farm and forest) per village were sampled over 4 nights. Each habitat was assigned to one team of two individuals. Sampling was restricted to human landing catches (HLC) and sampling was conducted between 6pm and midnight. Mosquitoes were collected into ethanol and identified, based on morphological characters, retrospectively. DNA was extracted from the ethanol preserved mosquitos and Plasmodium species were detected by PCR. 

It is not possible to comment on the integrity of DNA in the samples following ethanol preservation – however given the small number of Plasmodium vectors collected during the entire study this is perhaps a moot point. All sites were studied in three months (March to June 2016) the single time frame did not seem to particularly coincide with annual zoonotic malaria prevalence in Sabah. Even-so to attempt sampling 3 habitats, a team of two per habitat, using only HLC in 11 villages over a 3-month period is a long shot. 

While on the face of it this may appear to be an ‘intensive’ study the protocol and team size assigned for sampling at each habitat per site is, to say the least minimal, and lacks any attempt to include diverse sampling methods with no opportunity to identify differences in vector density over time. Furthermore, the study is unlikely to capture enough data to calculate vector association with seroprevalence data collected six months earlier.

Results

While data were analysed using formulae designed to detect vector prevalence the number of vectors collected at each habitat per site were small. Measures of diversity indicated that farming areas and the forest have a greater abundance of Anopheline vectors than the peri-domestic habitats. 

Conclusions

At best all that can be concluded is that the sampling methods used over short time intervals may not have been sufficient to properly capture the information required to calculate the environmental determinants of vector density and infection rate at the study sites during a single short time interval.

Malaria transmission in Southeast Asia, particularly Malaysian Borneo, is under intense control pressure and consequently is relatively low and temporal. The authors attempt to generate information on zoonotic malaria vector density, at best difficult, but they adopt scant, short and single capture methodologies that, in the best of circumstances, would be unlikely to generate the depth, quality or quantity of data required to identify the environmental determinants of Plasmodium knowlesi vector density and infections rates across a wider spatial scale in Sabah. 

 The authors do acknowledge this but yet make statements such as An. balabacensis occurred at the study sites at lower density when compared to a much more intense and prolonged study in the Kudat District of Sabah. Had the two studies been similarly rigorous then indeed a comparison could have been made. 

In the absence of rigorous longitudinal sampling the objectives of this study on the environmental determinants of malaria vector density, particularly zoonotic malaria, would be expected to be difficult.

Reviewer #2: This is a useful data set and data on vector species in this increasingly epidemiologically important region is scarce and notoriously difficult to collect. This makes the data presented here novel and significant to those currently attempting to assess a rapidly evolving transmission landscape. Overall, the vector sampling study is well designed and executed. The data are explained clearly and rigorously assessed. There are a few weaknesses in the study related to the use of serology data, but this weakness is clearly identified. With some additional evaluation of these weakness and discussion of alternative explanations for the lack of correlation observed here I feel the study will be ready for publication.

Reviewer #3: (No Response)

PLOS authors have the option to publish the peer review history of their article (what does this mean?). If published, this will include your full peer review and any attached files.

Reviewer #1: No

Reviewer #2: No

Reviewer #3: No
---

## [Decision Letter · Decision Letter 1]

1 Jul 2020

Dear Ms Brown,

Thank you very much for submitting your manuscript "Human exposure to zoonotic malaria vectors in village, farm and forest habitats in Sabah, Malaysian Borneo" for consideration at PLOS Neglected Tropical Diseases. As with all papers reviewed by the journal, your manuscript was reviewed by members of the editorial board and by several independent reviewers. The reviewers appreciated the attention to an important topic. Based on the reviews, we are likely to accept this manuscript for publication, providing that you modify the manuscript according to the review recommendations. 

Sincerely,

Hans-Peter Fuehrer

Deputy Editor

Reviewer's Responses to Questions

**Key Review Criteria Required for Acceptance?**

**Methods**

-Are the objectives of the study clearly articulated with a clear testable hypothesis stated?

-Is the study design appropriate to address the stated objectives?

-Is the population clearly described and appropriate for the hypothesis being tested?

-Is the sample size sufficient to ensure adequate power to address the hypothesis being tested?

-Were correct statistical analysis used to support conclusions?

-Are there concerns about ethical or regulatory requirements being met?

Reviewer #2: The revisions have improved the clarity of how the methods test the hypotheses. The study design is also clearer with clear and correct use of statistical methods.

Reviewer #3: (No Response)

**Results**

-Does the analysis presented match the analysis plan?

-Are the results clearly and completely presented?

-Are the figures (Tables, Images) of sufficient quality for clarity?

Reviewer #2: Yes. Where requested the authors have provided addition data and analyses. The addition of the power test is really good.

Reviewer #3: (No Response)

**Conclusions**

-Are the conclusions supported by the data presented?

-Are the limitations of analysis clearly described?

-Do the authors discuss how these data can be helpful to advance our understanding of the topic under study?

-Is public health relevance addressed?

Reviewer #2: Yes. The authors conclusions are well-justified by the data presented and they discuss potential limitations. The work certainly advances our understanding of vector ecology in this important area and can be used to inform public health investigation moving forward.

Reviewer #3: (No Response)

**Editorial and Data Presentation Modifications?**

Reviewer #2: No additional comments. The current version reads very clearly.

Reviewer #3: (No Response)

**Summary and General Comments**

Reviewer #2: The revision has addressed all of my concerns. The data set remains a rare and important contribution to the literature.

Reviewer #3: Please find in the following my comments on the revised manuscript titled, “Human exposure to zoonotic malaria vectors in village, farm and forest habitats in Sabah, Malaysian Borneo” by Brown et al. The authors revised the manuscript substantially.

Specific comments

1. L41: (2) is it a typographical error?

2. L73: SE should be South East.

3. L110: Please edit km. The meaning is unclear.

4. L132: What does “active” mean?

PLOS authors have the option to publish the peer review history of their article (what does this mean?). If published, this will include your full peer review and any attached files.

Reviewer #2: No

Reviewer #3: No
---

## [Editor Report · Decision Letter 2]

20 Jul 2020

Dear Ms Brown,

We are pleased to inform you that your manuscript 'Human exposure to zoonotic malaria vectors in village, farm and forest habitats in Sabah, Malaysian Borneo' has been provisionally accepted for publication in PLOS Neglected Tropical Diseases.

Best regards,

Hans-Peter Fuehrer

Deputy Editor

Hans-Peter Fuehrer

Deputy Editor

Please check Tab. 1 if gp. is correct (once gp. once in italics).